# Insecticide-treated net (ITN) use, factors associated with non-use of ITNs, and occurrence of sand flies in three communities with reported cases of cutaneous leishmaniasis in Ghana

Richard Akuffo [1,2,3]*, Michael Wilson[1], Bismark Sarfo[2], Phyllis Dako-Gyeke[2], Richard Adanu[2], Francis Anto[2]

1 Noguchi Memorial Institute for Medical Research, University of Ghana, Accra, Ghana, 2 School of Public Health, University of Ghana, Accra, Ghana, 3 University of Ghana Medical Centre, University of Ghana, Accra, Ghana

☯ These authors contributed equally to this work.
* richard.akuffo@gmail.com, rakuffo@ug.edu.gh

## Abstract

### Background

The insecticide treated bed net (ITN) has been proven for malaria control. Evidence from systematic review also suggests benefits of ITN roll out in reducing the incidence of cutaneous leishmaniasis (CL) and other vector borne diseases.

### Methods

Using a community-based cross-sectional study design, ITN use, factors associated with non-use of ITNs, and occurrence of sand flies were investigated in three communities with reported cases of CL in the Oti region of Ghana.

### Results

A total of 587 households comprising 189 (32.2%), 200 (34.1%), and 198 (33.7%) households from Ashiabre, Keri, and Sibi Hilltop communities with de facto population of 3639 participated in this study. The proportion of households that owned at least one ITN was 97.1%. The number of households having at least one ITN for every two members was 386 (65.8%) and 3159 (86.8%) household population had access to ITN. The household population that slept in ITN the night before this survey was 2370 (65.1%). Lack of household access to ITN (AOR = 1.80; CI: 1.31, 2.47), having a family size of more than 10 members (AOR = 2.53; CI: 1.20, 4.24), having more than 10 rooms for sleeping in a household (AOR = 10.18; CI: 1.28, 81.00), having 2–4 screened windows (AOR = 1.49; CI: 1.00, 2.20), and having 8–10 screened windows (AOR = 3.57; CI: 1.25, 10.17) were significantly associated with increased odds of not sleeping in ITN the night before the survey. A total of 193 female sand flies were trapped from various locations within the study communities.

**Data Availability Statement:** All relevant data are within the manuscript and its Supporting Information files.

**Funding:** This project was funded by the post graduate training scheme fellowship in implementation science program of the Special Program for Research and Training in Tropical Diseases (WHO/TDR) at the School of Public Health, University of Ghana. The funder had no role in the study design, data collection, data analysis, data interpretation, and in writing the manuscript.

**Competing interests:** The authors have declared that no competing interests exist.

## Conclusions

Factors associated with ITN non-use such as lack of household access to ITN should be incorporated into future efforts to improve ITN use. Species of sand flies and their potential vectorial role in the study communities should also be investigated.

## Introduction

Insecticide-treated nets (ITNs) are proven for malaria control and have played a significant role in reducing the global malaria burden by about two-thirds between 2000 and 2015 [1, 2]. Over the years, investments have been made into improving access to the ITNs and more people now own and use them than a few decades ago, especially in Africa. This may have contributed to the significant gains observed in the reduction of the global malaria burden. Some of the investments include free mass ITN distribution campaigns, ITN distribution at antenatal clinics and schools, among other measures [3–7].

The World Health Organization (WHO) defines universal coverage of ITN as "universal access to, and use of, ITNs by populations at risk of malaria" [2]. The minimum target for universal coverage to be considered achieved is usually 80% for both ITN access and use [8]

To measure ITN access and use, the Roll Back Malaria Monitoring and Evaluation Reference Group recommends the following four indicators: (i) the proportion of households that own at least one ITN, (ii) the proportion of households that own at least one ITN for two people, (iii) the proportion of the population with access to an ITN within the household, and (iv) the proportion of the population that used an ITN the previous night [8, 9].

Although improvements have been made over the years with these indicators, particularly ownership of at least one ITN by households, progress has been unequal across countries and communities, thereby requiring consistent monitoring of the indicators within various contexts [9, 10]

The Ghana national malaria control program actively promotes the use of ITNs for malaria control and aimed at reducing the malaria morbidity and mortality by 75.0% in its 2015–2020 Ghana malaria strategic plan. Some specific objectives in line with achieving the proposed reduction in malaria burden in Ghana include the following: 100% of households will own at least one ITN and 80% of the general population will sleep under ITNs [11, 12].

Vector control is also a key component of many anti-leishmaniasis programs and is likely to remain so until an effective vaccine against *Leishmania* infection becomes available. Some of the vector control methods used in the control of leishmaniasis include the ITN, insecticide impregnated durable wall lining (DWL), and indoor residual spraying [13–16]. Leishmaniasis is a neglected vector borne disease caused by parasites of the genus *Leishmania* and is endemic in over 98 countries with 350 million people estimated to be at risk of contracting the disease globally [17, 18].

Depending on the area of localization of the *Leishmania* parasite in mammalian tissues, two broad categories of leishmaniasis exist: visceral and cutaneous, with cutaneous leishmaniasis (CL) being the most common. Globally, it is estimated that between 0.7 to 1.3 million new cases of CL are reported every year [19, 20].

Leishmaniasis is geographically classified as New World or Old World depending on the distribution of the infecting *Leishmania* parasites. The New World species are usually found in Central and South America, whereas the Old World group is found in the Middle East, Asia, Africa, and the Mediterranean [21, 22]. Natural transmission of the *Leishmania* parasites to

humans and other mammals in the Old World occurs through the bite of various species of infected female phlebotomine sand flies belonging to the genus *Phlebotomus* [23–25]

Recent studies further suggest that ITNs may also be effective against other vector borne diseases (VBDs) such as CL. In this regard, a meta-analysis demonstrated a 77% reduction in the incidence of CL, attributable to ITN use. As a result, the roll out of ITN is particularly recommended in areas with high malaria and CL co-morbidities [26].

Cases of CL have been previously confirmed in the Ho municipality of the Volta Region of Ghana with many questions about the disease epidemiology such as vectors, reservoirs, and disease distribution still not fully answered [27–29].

In the year 2018, researchers at the Noguchi Memorial Institute for Medical Research involved in CL research in the Volta Region of Ghana, received reports about cases of skin ulcers which were suggestive of CL in some communities of the Oti region (which until 2019 was part of the Volta Region) [Naiki Attram personal communication].

This study was therefore conducted as part of a larger study investigating *Leishmania* infection and ITN use in three communities of the Oti region of Ghana, to obtain data on ITN use, factors associated with non-use of ITNs, and the occurrence of sand flies. The aspect of the larger study which investigated *Leishmania* infection in the study area confirmed exposure to *Leishmania* parasites by using the Leishmanin skin test (LST) and also detected cutaneous leishmaniasis among some of the persons with skin ulcers [30, 31].

## Materials and methods

### Ethics statement

Ethical approval to conduct this study was obtained from the ethics review committee of the Ghana Health Service (GHS-ERC006/08/18). Written informed consent was obtained from all study participants.

### Study design

Using a cross-sectional study design, this study was conducted in three communities of the Oti region of Ghana from October to December 2018. ITN ownership, access, use, and factors associated with non-use of ITN were investigated through a household survey. The occurrence of sand flies in the following locations of each study community was also investigated using CDC light traps (outdoor) and indoor aspiration: households, school, church, and mosque.

### Study area

This study was conducted in the following three communities of Ghana: Ashiabre, Keri, and Sibi Hilltop. Ashiabre is in the Tutukpene sub-district of the Nkwanta South municipality of the Oti Region of Ghana while Keri is in the Keri sub-district of the municipality. Sibi Hilltop is in the Sibi sub-district of the Nkwanta North district of the region. The climate of Ghana is tropical with two main seasons: Dry and wet seasons [32].

The population of Nkwanta South municipality is estimated to be 117,878 with males constituting 49.6% of the population. Covering a land area of approximately 2733 km$^2$, the Nkwanta South municipality is located between latitudes 7˚ 30' and 8˚ 45' North and longitude 0˚ 10' and 0˚ 45'East [33].

The population of the Nkwanta North district is estimated to be 64,553 with males constituting 50.2% of the population. The district is located between Latitude 7˚30'N and 8˚45'N and Longitude 0˚10'W and 045'E. It shares boundaries with Nkwanta South municipality to the

south, Nanumba South to the north, Republic of Togo to the east, and Kpandai District to the west [34].

## Inclusion criteria

Eligible study participants were household heads who were residents in the study community for $\geq$ 12 months.

For this study, a household was defined as a person or a group of persons, who live together in the same house or compound and share the same house-keeping arrangements. The head of each household was defined as a male or female member of the household recognised as such by the other household members. The head of a particular household is generally the person with economic and social responsibility for the household. As a result, household relationships were defined with reference the household head [35].

## Sample size consideration

To evaluate ownership, access and use of insecticide treated bed nets, a minimum of 475 households were required using the following formula and assumptions:

$$N = ((Z)^2 P/D^2) * (1-P)$$

Where, N = sample size, $Z^2$ = $(1.96)^2$ for 95% confidence interval (that is $\alpha$ = 0.05, P = proportion of household owning at least one ITN (75%), $D^2$ = maximum tolerable error for the prevalence estimate (0.05), design effect of 1.5 and a non-response rate of 10% [36–39].

## Selection of households for study inclusion

Using a sorted list of households, 200 households (with an average of 5–7 persons per household) were selected for study inclusion in each study community using a systematic sampling approach.

Details of household selection procedure for this study is published [31].

## Pre-study training

Prior to the commencement of field data collection, study team members were taken through a one-week training session comprising in-class training, break out discussion sessions, and field testing of the study questionnaires in a community in the Nkwanta South municipality (the main Nkwanta township).

The training sessions covered all aspects of the study procedures such as informed consent process and questionnaire administration. The field team comprised mainly of community-based volunteers.

## Household questionnaire administration

Using interviewer administered questionnaire, data on household ownership, access to and use of ITNs as well as factors which may be associated with non-use of ITN were obtained, with household heads as the respondents. The household heads also provided information on the number of household members, their relationship to each household member, educational level, age, and sex of all household members.

The household questionnaire also included questions on other household characteristics such as presence of electricity, main material of the household dwelling floor, main material of the roof, main material of the exterior wall of the household, number of rooms for sleeping,

household number of windows, number of windows with screen/net, place for cooking, main cooking fuel, main source of drinking water and main type of toilet facility.

The questionnaire also explored possessions of the households such as radio, television, telephone, and refrigerator. In addition, ownership of agricultural land and a means transportation such as bicycle, motorcycle and car were explored. Furthermore, information on specific characteristics of household heads such as religion, ethnicity, sex, and educational level was also obtained.

## Sand fly sampling in study communities

In a random sample of enrolled households (approximately 10 households per study community), sand fly collections were conducted for three consecutive nights in the sleeping area(s) of each household using battery powered indoor aspiration method for collection of resting flies from 4 am to 6 am each collection night. On the compound of the selected households, sand fly collections were conducted for three consecutive nights using battery powered CDC miniature light traps fitted with double ring fine mesh collection bags from 6 pm to 6 am each collection night (John W. Hock Company, Gainesville, FL).

Beyond the study households, sand flies were trapped outdoors using the CDC light trap at the following locations in each study community: a church compound, compound of a mosque and a school compound from 6 pm to 6 am each collection night for three consecutive nights. This was followed by indoor aspiration from 4 am to 6 am for each collection night for 3 consecutive nights. For each school selected, three classrooms were randomly selected one each from the nursery (KG), primary, and Junior high school departments for the indoor sand fly trapping using the aspiration method.

Sand flies collected were freeze-killed at -20˚C and sorted out into labeled 1.5ml eppendorf tubes containing silica gel for dry preservation. The tubes were secured in sealed Ziploc bags and transported to the entomology Laboratory at Noguchi Memorial Institute for Medical Research (NMIMR), University of Ghana, Legon-Accra. The sand flies were subsequently separated into either male or female based on morphology of their reproductive organ as was observed under a stereomicroscope (Olympus SZ60).

## Data management and analysis

Study data were captured using Microsoft Access software version 2013 and analyzed using STATA software version 14. Association between nominal variables in this study was assessed using Pearson's chi square test of association and Fishers exact test where cell counts below 5 were observed. Data analysis for this study was based on a 95% confidence level.

Using descriptive statistics, the following were determined:

**Proportion of households with at least one ITN.** This indicator was used to measure household ownership of an ITN. The numerator for this calculation was made up of all households having at least one ITN and the denominator was composed of the total of number of households.

**Proportion of households having a minimum of one ITN for every two household members.** This indicator was used to measure the proportion of households that had enough access to ITN (households having at least one ITN for every two household assuming that each ITN was used by two household members). To calculate this, the number of ITNs belonging to the household was divided by the number of individuals in the household. The numerator was made up of all households that had an ITN to people ratio of 0.5 or higher, while the denominator was the total number of households surveyed.

**Proportion of individuals with access to ITN within the households.** This indicator was used to estimate the proportion of study population that could use the existing ITNs, assuming that each ITN in a household was to be used by two people. The numerator was composed of all household members who had access to ITN in the study households, and the denominator was the de-facto population in the sample. Calculation of this indicator was done in two steps as outlined below.

First, an intermediate variable "potential ITN users" was calculated by multiplying the number of ITNs in each household by two. To adjust for households with more than one bed net for every two people, the potential ITN users were set equal to the members in that household if the potential users were more than the number of people in the household.

Next, the indicator for individual access was calculated by dividing the potential ITN users by the number of individuals in each household.

**Proportion of households with at least one ITN for every two people among households owning any LLIN.** This indicator measures the proportion of households owning at least one ITN and which had at least one ITN for every two members.

**Proportion of individuals who slept under ITNs the previous night.** This indicator measured the level of ITN use among all individuals at the time of the survey. The numerator was made up of all individuals who slept under an ITN the night prior to the survey, while the denominator was the total surveyed population.

**Ratio of ITN use to ITN access.** This indicator compared the indicator of individual ITN use to ITN access. This ratio is helpful in inferring whether the difference between ITN use and access could be explained as due to behavioral factors [40, 41].

For the ITN indicators analyzed, 95% confidence intervals (two sided) were estimated per study community and cumulatively. Binary (simple and multiple) logistic regression was used to estimate factors associated with failure to use the ITN.

Factors evaluated in the simple binary logistic regression for association with failure to use ITN included community of residence, household members' age, household members' sex, household members' educational level, sex of household head, age of household head, household size, main material in household roof, household number of rooms used for sleeping, number of windows in household, number of screened windows in household, whether household head heard any malaria message in the past 6 months, and household access to ITN.

Odds ratios for all variables included in the multiple logistic regression analysis with outcome being failure to use ITN the night before the survey, were adjusted for all covariates included in the model as well as for clustering at the household level using the vce (cluster clustvar) command in Stata statistical software version 14.

## Results and discussion

### Results

Of 600 households visited (200 from each study community), household heads from a total of 587 (97.8%) households comprising 189 (32.2%), 200 (34.1%), and 198 (33.7%) from Ashiabre, Keri and Sibi Hill Top respectively, were included in this study. The average household size was 6.3 with a range of 1 to 18 household members. Ashiabre and Sibi Hilltop had an average household size of 7 while Keri had an average household size of 5.

**Household head characteristics.** Table 1 summarizes key characteristics of the 587 household heads, with males constituting 82.8%. Majority were in the age categories of 31–40 years (32.5%) and 41–50 years (31.7%). Also, 429 (73.1%) of them had no formal education. Regarding religion of the household heads, 65 (11.1%) indicated that they did not belong to any religion, 323 (55.0%) were Christians, 176 (30.0%) were of traditional religion, and 23

**Table 1. Characteristics of the household heads.**

| Household head characteristics | Categories | Study Communities | | | |
|---|---|---|---|---|---|
| | | Ashiabre | Keri | Sibi Hill Top | Total |
| | | n (%) | n (%) | n (%) | n (%) |
| **Age (years)** | | | | | |
| | ≤ 20 | 1 (0.5) | 6(3.0) | 2 (1.0) | 9 (1.5) |
| | 21–30 | 18 (9.5) | 30 (15.0) | 25 (12.6) | 73 (12.4) |
| | 31–40 | 60 (31.8) | 65 (32.5) | 66 (33.3) | 191 (32.5) |
| | 41–50 | 59 (31.2) | 63 (31.5) | 64 (32.3) | 186 (31.7) |
| | 51–60 | 19 (10.1) | 23 (11.5) | 27 (13.6) | 69 (11.8) |
| | 61–70 | 15 (7.9) | 8 (4.0) | 9 (4.6) | 32 (5.5) |
| | ≥ 71 | 17 (9.0) | 5 (2.5) | 5 (2.5) | 27 (4.6) |
| **Sex** | | | | | |
| | Male | 164 (86.8) | 158 (79.0) | 164 (82.8) | 486 (82.8) |
| | Female | 25 (13.2) | 42 (21.0) | 34 (17.2) | 101 (17.2) |
| **Level of education** | | | | | |
| | No Formal Education | 131 (69.3) | 139 (69.5) | 159 (80.3) | 429 (73.1) |
| | Preschool | 7 (3.7) | 6 (3.0) | 4 (2) | 17 (2.9) |
| | Primary | 14 (7.4) | 22 (11.0) | 8 (4) | 44 (7.5) |
| | Junior High School | 21 (11.1) | 15 (7.5) | 16 (8.1) | 52 (8.9) |
| | Senior High School | 13 (6.9) | 15 (7.5) | 9 (4.5) | 37 (6.3) |
| | Tertiary | 3 (1.6) | 3 (1.5) | 2 (1.0) | 8 (1.4) |
| **Religion** | | | | | |
| | Catholic | 16 (8.5) | 76 (38) | 5 (2.5) | 97 (16.5) |
| | Protestant (Anglican, Presbyterian, Methodist, etc.) | 14 (7.4) | 10 (5.0) | 32 (16.2) | 56 (9.5) |
| | Pentecostal/Charismatic | 61 (32.3) | 33 (16.5) | 32 (16.2) | 126 (21.5) |
| | Other Christian | 13 (6.9) | 18 (9.0) | 13 (6.6) | 44 (7.5) |
| | Moslem | 13 (6.9) | 5 (2.5) | 5(2.5) | 23 (3.9) |
| | Traditional/Spiritualist | 47 (24.9) | 37 (18.5) | 92 (46.5) | 176 (30) |
| | No religion | 25 (13.2) | 21 (10.5) | 19 (9.6) | 65(11.1) |
| **Ethnicity** | | | | | |
| | Ewe | 1 (0.5) | 2 (1.0) | 0 (0) | 3 (0.5) |
| | Akan | 1 (0.5) | 12 (6.0) | 0 (0) | 13 (2.2) |
| | Mole-Dagbani | 1 (0.5) | 1 (0.5) | 2 (1.0) | 4 (0.7) |
| | Kokomba | 153 (81.0) | 27 (13.5) | 184 (92.9) | 364 (62.0) |
| | Grusi | 0 (0) | 2 (1.0) | 3 (1.5) | 5 (0.9) |
| | Achode | 0 (0) | 124 (62.0) | 0 (0) | 124 (21.1) |
| | Basare | 19 (10.1) | 1 (0.5) | 6 (3.0) | 26 (4.4) |
| | Challa | 0 (0) | 22 (11.0) | 0 (0) | 22 (3.7) |
| | Other | 14 (7.4) | 9 (4.5) | 3 (1.5) | 26 (4.4) |
| | **Total** | 189 (100) | 200 (100) | 198 (100) | 587 (100) |

(3.9%) were Muslims. The Christians were composed of the following: Catholics (97), Protestants (56), Pentecostals (126), and other Christians (44).

Although various ethnic groups were recorded among the heads of households, certain ethnic groups were more dominant in the respective study communities. In Ashiabre for instance, 81% of the household heads belonged to the Kokomba ethnic group while in Keri, the Achode ethnic group (62.0%) dominated. In Sibi Hilltop, it was observed that 92.9% of the household heads were members of the Kokomba ethnic group (Table 1).

**Table 2. Household composition by the number of usual household members, educational level, and relationship to head of household.**

| Household characteristics | Categories | Study Communities | | | |
|---|---|---|---|---|---|
| | | Ashiabre | Keri | Sibi Hill Top | Total |
| | | n (%) | n (%) | n (%) | n (%) |
| Number of household members | | | | | |
| | 1–3 persons | 26(2.0) | 113 (11.3) | 33 (2.4) | 172 (4.6) |
| | 4–6 persons | 441 (33.4) | 582 (58.0) | 402 (28.9) | 1425 (38.3) |
| | 7–9 persons | 467 (35.4) | 240 (23.9) | 565 (40.6) | 1272 (34.2) |
| | ≥10 persons | 387 (29.3) | 69 (6.9) | 393 (28.2) | 849 (22.8) |
| | Subtotal | 1321 (100) | 1004(100) | 1393 (100) | 3718 (100) |
| | Mean size of households | 7.0 | 5.0 | 7.0 | 6.3 |
| | Minimum household size | 2 | 1 | 2 | 1 |
| | Maximum household size | 18 | 13 | 16 | 18 |
| Educational Level | | | | | |
| | No Formal Education | 648 (49.1) | 549 (54.7) | 718 (51.5) | 1915 (51.5) |
| | Preschool | 159 (12) | 75 (7.5) | 139 (10) | 373 (10) |
| | Primary | 370 (28) | 285 (28.4) | 425 (30.5) | 1080 (29) |
| | Junior High School | 101 (7.6) | 64 (6.4) | 87 (6.2) | 252 (6.8) |
| | Senior High School | 35 (2.6) | 24 (2.4) | 22 (1.6) | 81 (2.2) |
| | Tertiary | 8 (0.6) | 7 (0.7) | 2 (0.1) | 17 (0.5) |
| Relationship to head of household | | | | | |
| | Head of household | 189 (14.3) | 200 (19.9) | 198 (14.2) | 587 (15.8) |
| | Wife/Husband | 189 (14.3) | 160 (15.9) | 210 (15.1) | 559 (15.0) |
| | Son/Daughter | 844 (63.9) | 555 (55.3) | 886 (63.6) | 2285 (61.5) |
| | Son-in-law/Daughter-in-law | 10 (0.8) | 10 (1.0) | 19 (1.4) | 39 (1.0) |
| | Grandchild | 28 (2.1) | 36 (3.6) | 26 (1.9) | 90 (2.4) |
| | Parent | 9 (0.7) | 3 (0.3) | 15 (1.1) | 27 (0.7) |
| | Parent-in-law | 3 (0.2) | 3 (0.3) | 8 (0.6) | 14 (0.4) |
| | Brother/Sister | 20 (1.5) | 11 (1.1) | 17 (1.2) | 48 (1.3) |
| | Brother-in-law/sister-in-law | 3 (0.2) | 2 (0.2) | 2 (0.1) | 7 (0.2) |
| | Uncle/Aunt | 4 (0.3) | 11 (1.1) | 2 (0.1) | 17 (0.5) |
| | Niece/ Nephew | 14 (1.1) | 4 (0.4) | 3 (0.2) | 21 (0.6) |
| | Other relative | 7 (0.5) | 5 (0.5) | 2 (0.1) | 14 (0.4) |
| | Adopted /Foster/ stepchild | 1 (0.1) | 4 (0.4) | 5 (0.4) | 10 (0.3) |
| Total | | 1321 (100) | 1004 (100) | 1,393.00 | 3718 (100) |

**Household composition.** In Ashiabre, 35.4% of the household members were from households having 7–9 individuals. This was closely followed by 33.4% and 29.3% of households with 4–6 persons, and ten or more persons, respectively. In Keri, the majority (58.0%) lived in households with 4–6 persons while in Sibi Hilltop, the majority (40.6%) lived in households with 7–9 members (Table 2).

Regarding educational level of household members, 51.5%, 54.7%, and 51.5% of the household members in Ashiabre, Keri, and Sibi Hilltop, respectively, had no formal education (Table 2). The overall proportion of the household members with tertiary level education in the study area was 0.5% (Table 2).

The majority of all the household members (61.5%) were children of the household heads. Other household members included sons-in-law/daughters-in-law, grandchildren, parents, parents-in-law, brothers/sisters, brothers-in-law/sisters-in-law, uncles/aunt, nieces/nephews, other relatives, and adopted/foster/stepchildren (Table 2).

**Table 3. Summary of household population distribution by sex, age, and community of residence.**

| | Ashiabre | | | Keri | | | Sibi Hill Top | | | Total | | |
|---|---|---|---|---|---|---|---|---|---|---|---|---|
| | Male | Female | Total | Male | Female | Total | Male | Female | Total | Male | Female | Total |
| Age (years) | n (%) | n (%) | n (%) | n (%) | n (%) | n (%) | n (%) | n (%) | n (%) | n (%) | n (%) | n (%) |
| <5 | 99 (14.7) | 90 (13.9) | 189(14.3) | 62 (12.4) | 62 (12.4) | 124(12.4) | 82 (11.4) | 103(15.3) | 185(13.3) | 243(12.8) | 255 (14) | 498(13.4) |
| 5–9 | 136(20.2) | 106(16.4) | 242(18.3) | 121(24.1) | 102(20.3) | 223(22.2) | 171(23.8) | 125(18.5) | 296(21.2) | 428(22.6) | 333(18.3) | 761(20.5) |
| 10–14 | 137(20.4) | 138(21.3) | 275(20.8) | 102(20.3) | 84 (16.7) | 186(18.5) | 181(25.2) | 136(20.2) | 317(22.8) | 420(22.2) | 358(19.6) | 778(20.9) |
| 15–19 | 90 (13.4) | 60 (9.3) | 150(11.4) | 35 (7) | 40 (8) | 75 (7.5) | 78 (10.8) | 54 (8) | 132 (9.5) | 203(10.7) | 154 (8.4) | 357 (9.6) |
| 20–24 | 28 (4.2) | 32 (4.9) | 60 (4.5) | 14 (2.8) | 28 (5.6) | 42 (4.2) | 25 (3.5) | 23 (3.4) | 48 (3.4) | 67 (3.5) | 83 (4.6) | 150 (4) |
| 25–29 | 16 (2.4) | 36 (5.6) | 52 (3.9) | 24 (4.8) | 27 (5.4) | 51 (5.1) | 21 (2.9) | 42 (6.2) | 63 (4.5) | 61 (3.2) | 105 (5.8) | 166 (4.5) |
| 30–34 | 20 (3) | 42 (6.5) | 62 (4.7) | 20 (4) | 42 (8.4) | 62 (6.2) | 34 (4.7) | 62 (9.2) | 96 (6.9) | 74 (3.9) | 146 (8) | 220 (5.9) |
| 35–39 | 39 (5.8) | 46 (7.1) | 85 (6.4) | 31 (6.2) | 34 (6.8) | 65 (6.5) | 28 (3.9) | 44 (6.5) | 72 (5.2) | 98 (5.2) | 124 (6.8) | 222 (6) |
| 40–44 | 20 (3) | 42 (6.5) | 62 (4.7) | 27 (5.4) | 37 (7.4) | 64 (6.4) | 33 (4.6) | 43 (6.4) | 76 (5.5) | 80 (4.2) | 122 (6.7) | 202 (5.4) |
| 45–49 | 32 (4.8) | 26 (4) | 58 (4.4) | 30 (6) | 24 (4.8) | 54 (5.4) | 22 (3.1) | 23 (3.4) | 45 (3.2) | 84 (4.4) | 73 (40) | 157 (4.2) |
| 50–54 | 19 (2.8) | 8 (1.2) | 27 (2) | 22 (4.4) | 9 (1.8) | 31 (3.1) | 19 (2.6) | 7 (1) | 26 (1.9) | 60 (3.2) | 24 (1.3) | 84 (2.3) |
| 55–59 | 5 (0.7) | 4 (0.6) | 9 (0.7) | 3 (0.6) | 5 (1) | 8 (0.8) | 6 (0.8) | 8 (1.2) | 14 (1) | 14 (0.7) | 17 (0.9) | 31 (0.8) |
| 60–64 | 8 (1.2) | 8 (1.2) | 16 (1.2) | 3 (0.6) | 4 (0.8) | 7 (0.7) | 9 (1.3) | 2 (0.3) | 11 (0.8) | 20 (1.1) | 14 (0.8) | 34 (0.9) |
| 65–69 | 7 (1) | 0 (0) | 7 (0.5) | 1 (0.2) | 0 (0) | 1 (0.1) | 2 (0.3) | 1 (0.1) | 3 (0.2) | 10 (0.5) | 1 (0.1) | 11 (0.3) |
| 70–74 | 5 (0.7) | 2 (0.3) | 7 (0.5) | 4 (0.8) | 1 (0.2) | 5 (0.5) | 2 (0.3) | 0 (0) | 2 (0.1) | 11 (0.6) | 3 (0.2) | 14 (0.4) |
| 75–79 | 2 (0.3) | 0 (0) | 2 (0.2) | 2 (0.4) | 1 (0.2) | 3 (0.3) | 4 (0.6) | 0 (0) | 4 (0.3) | 8 (0.4) | 1 (0.1) | 9 (0.2) |
| >80 | 10 (1.5) | 8 (1.2) | 18 (1.4) | 1 (0.2) | 2 (0.4) | 3 (0.3) | 2 (0.3) | 1 (0.1) | 3 (0.2) | 13 (0.7) | 11 (0.6) | 24 (0.6) |
| **Total** | **673 (100)** | **648 (100)** | **1321(100)** | **502 (100)** | **502 (100)** | **1004 (100)** | **719 (100)** | **674 (100)** | **1393 (100)** | **1894(100)** | **1824(100)** | **3718(100)** |

The detailed distribution of study household population by age, sex, and community of residence is presented in Table 3. Of the 3718 usual household members, 1894 (50.9%) made up of 673 (50.9%), 502 (50.0%), and 719 (51.6%) of the participants at Ashiabre, Keri and Sibi Hilltop respectively were males. Four hundred and ninety-eight (13.4%) of them were children under 5 years, and 2394 (64.4%) were less than 20 years old (Table 3). Data on additional household characteristics including household possessions is included as a (S1 File).

**ITN ownership and access.** Insecticide-treated nets owned by the households ranged from 1 to 13 with majority of them having 3 (35.9%) ITNs (Table 4). In the study communities,

**Table 4. Number of ITNs owned by households.**

| | Ashiabre | | Keri | | Sibi Hilltop | | Total | |
|---|---|---|---|---|---|---|---|---|
| ITN number | Households owning ITN, n(%) | Number of ITNs owned | Households owning ITN, n(%) | Number of ITNs owned | Households owning ITN, n(%) | Number of ITNs owned | Households owning ITN, n(%) | Number of ITNs owned |
| 0 | 5 (2.6) | 0 | 6 (3.0) | 0 | 6 (3.0) | 0 | 17 (2.9) | 0 |
| 1 | 4 (2.1) | 4 | 19 (9.5) | 19 | 10 (5.1) | 10 | 33 (5.6) | 33 |
| 2 | 33 (17.5) | 66 | 53 (26.5) | 106 | 29 (14.6) | 58 | 115 (19.6) | 230 |
| 3 | 59 (31.2) | 177 | 87 (43.5) | 261 | 65 (32.8) | 195 | 211 (35.9) | 633 |
| 4 | 30 (15.9) | 120 | 23 (11.5) | 92 | 25 (12.6) | 100 | 78 (13.3) | 312 |
| 5 | 28 (14.8) | 140 | 9 (4.5) | 45 | 27 (13.6) | 135 | 64 (10.9) | 320 |
| 6 | 14 (7.4) | 84 | 2 (1.0) | 12 | 15 (7.6) | 90 | 31 (5.3) | 186 |
| 7 | 9 (4.8) | 63 | 1 (0.5) | 7 | 14 (7.1) | 98 | 24 (4.1) | 168 |
| 8 | 5 (2.6) | 40 | 0 (0) | 0 | 6 (3.0) | 48 | 11 (1.9) | 88 |
| 9 | 1 (0.5) | 9 | 0 (0) | 0 | 0 (0) | 0 | 1 (0.2) | 9 |
| 10 | 0 (0) | 0 | 0 (0) | 0 | 1 (0.5) | 10 | 1 (0.2) | 10 |
| 13 | 1 (0.5) | 13 | 0 (0) | 0 | 0 (0) | 0 | 1 (0.2) | 13 |
| **Total** | **189 (100)** | **716** | **200 (100)** | **542** | **198 (100)** | **744** | **587 (100)** | **2002** |

**Table 5. Source, duration of ownership, and observation of bed nets owned by households.**

| Characteristic | Category | Ashiabre | | Keri | | Sibi Hilltop | | Total | |
|---|---|---|---|---|---|---|---|---|---|
| | | no. | % | no. | % | no. | % | no. | % |
| Source of bed net | | | | | | | | | |
| | Public Sector | 713 | 99.6 | 537 | 99.1 | 744 | 100 | 1994 | 99.6 |
| | Other/Don't know | 3 | 0.4 | 5 | 0.9 | 0 | 0 | 8 | 0.4 |
| Duration of bed net ownership | | | | | | | | | |
| | 0–6 months | 430 | 60.1 | 450 | 83.0 | 373 | 50.1 | 1253 | 62.6 |
| | 7–12 months | 279 | 39.0 | 85 | 15.7 | 356 | 47.8 | 720 | 36.0 |
| | Not sure | 7 | 0.9 | 7 | 1.3 | 15 | 2.0 | 29 | 1.4 |
| Bed net observation | | | | | | | | | |
| | Observed Hanging | 409 | 57.1 | 387 | 71.4 | 430 | 57.8 | 1226 | 61.2 |
| | Observed Not Hanging or packaged | 271 | 37.9 | 108 | 19.9 | 238 | 32 | 617 | 30.8 |
| | Not observed | 36 | 5.0 | 47 | 8.7 | 76 | 10.2 | 159 | 7.9 |
| **Total bed nets owned** | | **716** | **100** | **542** | **100** | **744** | **100** | **2002** | **100** |

59 (31.2%), 87 (43.5%), and 65 (32.8%) households owned 3 ITNs in Ashiabre, Keri, and Sibi Hilltop, respectively. Cumulatively, the study households owned 2002 ITNs distributed as follows: 716 in Ashiabre, 542 in Keri, and 744 in Sibi Hilltop (Table 4).

Regarding cost of the ITNs, all respondents indicated that they obtained the ITNs at no financial cost to them (free of charge). All respondents also indicated that they had heard about malaria. Most (99.6%) of the ITNs were obtained from the public sector sources such as the Government hospital, health post/CHPS compound, and national ITN distribution campaigns (Table 5). Of the ITNs owned, 1253 (62.6%) and 1973 (98.6%) were obtained within 6 months and 12 months of this study initiation respectively (Table 5).

Furthermore, 1226 (61.2%) of the ITNs owned were observed hanging, 617 (30.8%) were either not hanging or packaged, while 159 (7.9%) were not observed. A similar trend was observed across the individual study communities (Table 5).

Within a period of 12 months prior to this study, respondents indicated that in 496 (84.6) of the 587 households, at least one ITN had been disposed of using different methods. Across the study communities, the commonest methods of ITN disposal were garbage /refuse dump (45.2%) followed by burning (42.3%) (Table 6). Regarding duration of bed net use before disposal, majority of the household heads (69.4%) indicated that the nets had been used for a period of 2–4 years prior to disposal. This was followed by 121 (24.4%) household heads who indicated that their bed nets had been used for periods less than 2 years prior to disposal (Table 6).

Among the reasons for bed net disposal, 404 (81.5) respondents indicated that their nets were disposed of because they were torn. This was followed by 55 (11.1%) who indicated that their bed nets were disposed of because they had obtained a new one (Table 6).

The proportion of households with at least one insecticide-treated net (this case the long-lasting insecticidal net (LLIN)) was 97.1% (95% CI: 95.4, 98.2). In both Keri and Sibi Hilltop, 97.0% of the study households owned at least one ITN with 97.4% in Ashiabre (Table 7).

Cumulatively, 386 (65.8%) households owned at least one ITN for every two household members. The proportion of households with at least one ITN for every two household members was 63.5%, 68.0%, and 65.7% in Ashiabre, Keri, and Sibi Hilltop respectively (Table 7).

Furthermore, the overall proportion of the individuals that could be potentially covered by the existing ITNs, if each ITN in the household could be used by two people (proportion of individuals with access to ITN within the households) was estimated as 86.8% (95% CI: 85.7,

**Table 6. Methods of ITN disposal, duration of ITN use before disposal, and reason for ITN disposal.**

| Characteristics | Category | Ashiabre | | Keri | | Sibi Hilltop | | Total | |
|---|---|---|---|---|---|---|---|---|---|
| | | no. | % | no. | % | no. | % | No. | % |
| Method of treated net disposal | | | | | | | | | |
| | Burned | 44 | 30.3 | 88 | 48.9 | 78 | 45.6 | 210 | 42.3 |
| | Buried | 2 | 1.4 | 12 | 6.7 | 10 | 5.8 | 24 | 4.8 |
| | Garbage or refuse dump | 87 | 60.0 | 69 | 38.3 | 68 | 39.8 | 224 | 45.2 |
| | Reused for other purpose | 9 | 6.2 | 8 | 4.4 | 14 | 8.2 | 31 | 6.3 |
| | Other | 3 | 2.1 | 3 | 1.7 | 1 | 0.6 | 7 | 1.4 |
| How long was treated net used before disposing of it? | | | | | | | | | |
| | Less than 2 years | 27 | 18.6 | 43 | 23.9 | 51 | 29.8 | 121 | 24.4 |
| | 2–4 years | 102 | 70.3 | 134 | 74.4 | 108 | 63.2 | 344 | 69.4 |
| | More than 4 years | 11 | 7.6 | 2 | 1.1 | 10 | 5.8 | 23 | 4.6 |
| | Don't know | 5 | 3.4 | 1 | 0.6 | 2 | 1.2 | 8 | 1.6 |
| What was the main reason for disposing of the treated net? | | | | | | | | | |
| | Torn | 114 | 78.6 | 148 | 82.2 | 142 | 83 | 404 | 81.5 |
| | Could not repel mosquitoes anymore | 6 | 4.1 | 17 | 9.4 | 9 | 5.3 | 32 | 6.5 |
| | Got a new one | 22 | 15.2 | 14 | 7.8 | 19 | 11.1 | 55 | 11.1 |
| | Other/Don't Know | 3 | 2.1 | 1 | 0.6 | 1 | 0.6 | 5 | 1.0 |
| Households in which any treated net was disposed of in the past 12 months | | | | | | | | | |
| | Total | 145 | 100 | 180 | 100 | 171 | 100 | 496 | 100 |

87.9) with similar proportions observed in the individual study communities (87.3%, 87.9%, and 85.6% in Ashiabre, Keri, and Sibi Hilltop respectively) (Table 8).

**ITN use.** The overall proportion of the study participants that used ITNs the night before the interview was 65.1% (95% CI: 63.6, 66.7). In the respective study communities, the proportions were 66.4% (95% CI: 63.7, 68.9), 65.1% (95% CI: 62.1, 68.0), and 64.0% (95% CI: 61.4, 66.5) in Ashiabre, Keri and Sibi Hilltop, respectively. The overall ratio of ITN use to ITN access observed was 0.75 (Table 8).

In addition, it was observed among households having at least one ITN for every two family members that 1,581 (72.5%) of the household members slept in an ITN the previous night, with similar proportions of 73.5%, 71.1%, and 72.6 observed for same group in Ashiabre, Keri, and Sibi Hilltop respectively (Table 9).

A summary of the distribution of persons who slept in the ITN the night before the study by age, sex, and community of residence is presented in Table 10 below with 1197 (50.5%) of them being males. Majority of ITN users were within the age groups of 5–15 years (42.1%) and 16–45 years (35.6%) respectively.

**Table 7. Ownership of ITNs by enrolled households in study community.**

| Study Community | Households Interviewed | Households with at least one ITN, n (%) | | Households with at least one ITN for every two people | |
|---|---|---|---|---|---|
| | | n (%) | 95% CI | n (%) | 95% CI |
| Ashiabre | 189 | 184 (97.4) | (93.7, 98.9) | 120 (63.5) | (56.3, 70.1) |
| Keri | 200 | 194 (97.0) | (93.4, 98.7) | 136 (68.0) | (61.2, 74.1) |
| Sibi Hilltop | 198 | 192 (97.0) | (93.4, 98.6) | 130 (65.7) | (58.7, 72.0) |
| Total | 587 | 570 (97.1) | (95.4, 98.2) | 386 (65.8) | (61.8, 69.5) |

**Table 8. Access to and use of LLINs by enrolled households in study communities.**

| Study Community | Household Population (de facto) | Population with access to ITN within their household | | Population that slept in ITN the night prior to the study | | Ratio of use to access |
|---|---|---|---|---|---|---|
| | | n (%) | 95% CI | n (%) | 95% CI | |
| Ashiabre | 1279 | 1116 (87.3) | (85.3, 89.0) | 849(66.4) | (63.7, 68.9) | 0.76 |
| Keri | 983 | 864 (87.9) | (85.7, 89.8) | 640 (65.1) | (62.1, 68.0) | 0.74 |
| Sibi Hilltop | 1377 | 1179 (85.6) | (83.7, 87.4) | 881 (64.0) | (61.4, 66.5) | 0.75 |
| Total | 3639 | 3159 (86.8) | (85.7, 87.9) | 2370(65.1) | (63.6, 66.7) | 0.75 |

Among females, majority of ITN users were 16–45 years old (42.4%) and 5–15 years old (36.8%) respectively. Among males, majority of ITN users (47.3%) were 5–15 years old and 16–45 years old (28.9%) respectively (Table 10).

**Factors associated with non-use of ITNs.** This study also observed significant associations between not sleeping under ITN the night before the household survey and the following factors using multiple logistic regression: family size and number of rooms used for sleeping (Table 11). Additional factors found to be significantly associated with failure to sleep under ITN the night before this survey were number of screened windows in household, and household lacking access to ITN (Table 11).

Participants from households with size of 10 or more members (AOR = 2.53; 95% CI: 1.20, 4.24) were more likely not to use ITN than those from households with size less than 10. Participants from households having >10 rooms for sleeping (AOR = 10.18; 95% CI: 1.28, 81.0) had greater odds of not using ITN than those from households having <10 rooms for sleeping. In addition, participants from households having 2–4 screened windows (AOR = 1.49; 95% CI: 1.00, 2.20), and 8–10 screened windows (AOR = 3.57; 95% CI: 1.25, 10.17) had higher likelihood of not using ITN compared with those not having screened windows.

Participants from households which did not have one ITN for every two household members (AOR = 1.80;95% CI: 1.31, 2.47) had higher odds of failing to use ITN compared with participants from households which had at least one ITN for every two household members. Existence of CL history in family or exposure to *Leishmania* infection measured by the leishmanin skin test (LST) was not associated with increased odds of not sleeping in ITN.

**Occurrence of sand flies in study communities.** A total of 218 sand flies comprising of 25 males and 193 females were trapped using both the CDC light trap and indoor aspiration methods. Of the 193 female sand flies, 165 were trapped using the CDC light trap while 28 were trapped using the indoor aspiration method. Of the 165 female sand flies trapped using the CDC light traps, 131 (79.4%) were trapped from household compounds. In addition, 7 (4.2%), 25 (15.2%), and 2 (1.2%) were trapped from church compound, school compound, and mosque compound respectively (Table 12).

**Table 9. Use of ITNs by members of households having a minimum of one ITN for every two members.**

| Study Community | Persons in Households with ITN access (de facto) | Persons that used ITN the night prior to the study among households with ITN access | |
|---|---|---|---|
| | | n(%) | 95% CI |
| Ashiabre | 736 | 541(73.5) | (70.2, 76.6) |
| Keri | 620 | 441 (71.1) | (67.4, 74.6) |
| Sibi Hilltop | 825 | 599 (72.6) | (69.5, 75.5) |
| Total | 2181 | 1581(72.5) | (70.6, 74.3) |

**Table 10. Distribution of persons who used ITN the night before the survey by sex, age group and residence.**

| Community | Age groups | Male | Female | Total | p-value |
|---|---|---|---|---|---|
| Ashiabre | < 5 years | 65 (14.8) | 65 (15.9) | 130 (15.3) | 0.002 |
| | 5–15 years | 198 (45.1) | 150 (36.6) | 348 (41.0) | |
| | 16–45 years | 127 (28.9) | 165 (40.2) | 292 (34.4) | |
| | >45 years | 49 (11.2) | 30 (7.3) | 79 (9.3) | |
| | Sub total | 439 (100) | 410 (100) | 849 (100) | |
| Keri | < 5 years | 38 (12.3) | 37 (11.2) | 75 (11.7) | 0.002 |
| | 5–15 years | 135 (43.5) | 114 (34.5) | 249 (38.9) | |
| | 16–45 years | 96 (31.0) | 149 (45.2) | 245 (38.3) | |
| | >45 years | 41 (13.2) | 30 (9.1) | 71 (11.1) | |
| | Sub total | 310 (100) | 330 (100) | 640 (100) | |
| Sibi Hilltop | < 5 years | 52 (11.6) | 59 (13.6) | 111 (12.6) | <0.001 |
| | 5–15 years | 233 (52.0) | 168 (38.8) | 401 (45.5) | |
| | 16–45 years | 123 (27.5) | 183 (42.3) | 306 (34.7) | |
| | >45 years | 40 (8.9) | 23 (5.3) | 63 (7.2) | |
| | Sub total | 448 (100) | 433 (100) | 881 (100) | |
| Total | < 5 years | 155 (12.9) | 161 (13.7) | 316 (13.3) | <0.001 |
| | 5–15 years | 566 (47.3) | 432 (36.8) | 998 (42.1) | |
| | 16–45 years | 346 (28.9) | 497 (42.4) | 843 (35.6) | |
| | >45 years | 130 (10.9) | 83 (7.1) | 213 (9.0) | |
| | Total | 1197 (100) | 1173(100) | 2370 (100) | |

Of the 28 female sand flies caught using indoor aspiration, 9(32.1%), 2(7.1%), 3(10.7%), 3 (10.7%), and 11(39.3%) were collected from Junior High School classroom, KG classroom, primary classroom, inside church, and household sleeping area respectively (Table 12).

## Discussion

**Insecticide treated bed net ownership and access.** This study investigated ITN ownership and use, and was conducted as part of a larger study which established the prevalence of cutaneous leishmaniasis (CL) in the study communities [30]. Results from this study indicate that 97.1% of households surveyed owned at least one ITN and 86.8% of the study population had access to ITN. Between 2016 and 2018, Ghana was one of a total of eight countries that received 50% of the global distribution of ITNs. Evaluation of ITN indicators in Ghanaian communities is therefore important in providing feedback to inform future improvements of the intervention delivery [42].

Over the years, several countries including Ghana have made significant strides in increasing the number of households that own ITNs through the adoption of several ITN (intervention) delivery strategies such as mass ITN distribution campaigns and continues distribution of ITNs during antenatal clinics and other delivery channels [6, 7, 10].

The observations made regarding ITN ownership and proportion of household population having access to ITN in the study communities are improvements over what was observed in the Volta region during the 2014 Ghana demographic and health survey, the 2019 GMIS, as well as the individual studies in the Volta Region cited above. Current regional level estimates of the new Oti region for the ITN indicators discussed above will be helpful in comparing the observations in the study communities [43, 44].

Evaluating access to ITN at both the household and individual levels is important in explaining the ITN use observed. The fact that a household has at least one ITN may not mean

**Table 11. Factors associated with non-use of ITN the night preceding the interview among de facto population of households having a minimum of one ITN.**

| Characteristics | Categories | defacto population | People that did not sleep under ITN, n(%) | Crude OR (95% CI) | P value | *AOR (95% CI) | P value |
|---|---|---|---|---|---|---|---|
| Sex of Household head | | | | | | | |
| | Female | 452 | 115 (25.4) | [Reference] | | [Reference] | |
| | Male | 3095 | 1062 (34.3) | 1.53 (1.22, 1.92) | <0.001 | 1.45 (0.96, 2.20) | 0.077 |
| Household head age | | | | | | | |
| | <35 years | 820 | 240 (29.3) | [Reference] | | [Reference] | |
| | 36–40 years | 610 | 171 (28.0) | 0.94 (0.75, 1.19) | 0.610 | 0.98 (0.63,1.51) | 0.910 |
| | 41–50 years | 1226 | 443 (36.1) | 1.37 (1.13, 1.65) | 0.001 | 1.24 (0.84,1.84) | 0.275 |
| | >51 years | 891 | 323 (36.3) | 1.37 (1.12, 1.68) | 0.002 | 1.15 (0.76, 1.73) | 0.515 |
| Family size | | | | | | | |
| | 1–3 persons | 157 | 33 (21.0) | [Reference] | | [Reference] | |
| | 4–6 persons | 1380 | 374 (27.1) | 1.40 (0.93, 2.09) | 0.103 | 1.15 (0.81, 2.24) | 0.249 |
| | 7–9 persons | 1203 | 404 (33.6) | 1.90 (1.27, 2.84) | 0.002 | 1.34 (0.83, 2.65) | 0.182 |
| | > = 10 persons | 807 | 366 (45.4) | 3.12 (2.07, 4.69) | <0.001 | 2.53 (1.20, 4.24) | 0.011* |
| Main material in household roof | | | | | | | |
| | Metal | 3234 | 1057 (32.7) | [Reference] | | [Reference] | |
| | Thatch | 313 | 120 (38.3) | 1.28 (1.00, 1.63) | 0.043 | 1.12 (0.70, 1.78) | 0.643 |
| Household socioeconomic status | | | | | | | |
| | Low | 658 | 199 (30.2) | [Reference] | | [Reference] | |
| | Second | 735 | 237 (32.2) | 1.10 (0.87, 1.38) | 0.421 | 1.09 (0.65, 1.82) | 0.755 |
| | Middle | 758 | 259 (34.2) | 1.20 (0.96, 1.50) | 0.115 | 1.22 (0.70, 2.10) | 0.481 |
| | Fourth | 781 | 259 (33.2) | 1.14 (0.92, 1.43) | 0.236 | 1.15 (0.66, 2.02) | 0.624 |
| | Highest | 615 | 223 (36.3) | 1.31 (1.04, 1.66) | 0.023 | 1.39 (0.85, 2.27) | 0.188 |
| Household number of rooms for sleeping | | | | | | | |
| | 1 room | 259 | 70 (27.0) | [Reference] | | [Reference] | |
| | 2–5 rooms | 2553 | 802 (31.4) | 1.23 (0.93, 1.65) | 0.146 | 1.30 (0.35, 4.75) | 0.693 |
| | 6–10 rooms | 613 | 253 (41.3) | 1.90 (1.38, 2.61) | <0.001 | 2.29 (0.60, 8.77) | 0.225 |
| | >10 rooms | 122 | 52 (42.6) | 2.01 (1.28, 3.15) | 0.003 | 10.18 (1.28, 81.00) | 0.028* |
| Number of windows in household | | | | | | | |
| | 1 window | 276 | 81 (29.4) | [Reference] | | [Reference] | |
| | 2–4 windows | 2100 | 662 (31.5) | 1.11 (0.84, 1.46) | 0.464 | 0.74 (0.20, 2.79) | 0.656 |
| | 5–7 windows | 847 | 312 (36.8) | 1.40 (1.05, 1.88) | 0.024 | 0.66 (0.17, 2.47) | 0.533 |
| | 8–10 windows | 231 | 96 (41.6) | 1.71 (1.18, 2.47) | 0.004 | 0.41 (0.09, 1.77) | 0.232 |
| | >10 windows | 93 | 26 (28.0) | 0.93 (0.55, 1.57) | 0.798 | 0.06 (0.01, 0.57) | 0.015 |
| Number of screened windows in household | | | | | | | |
| | No screened window | 2476 | 765 (30.9) | [Reference] | | [Reference] | |
| | 1 screened window | 223 | 59 (26.5) | 0.80 (0.59, 1.10) | 0.169 | 0.76 (0.41, 1.41) | 0.387 |
| | 2–4 screened windows | 557 | 208 (37.3) | 1.33 (1.10, 1.61) | 0.003 | 1.49 (1.00, 2.20) | 0.047* |
| | 5–7 screened windows | 190 | 92 (48.4) | 2.10 (1.56, 2.83) | <0.001 | 1.68 (0.85, 3.32) | 0.136 |
| | 8–10 screened windows | 96 | 53 (55.2) | 2.76 (1.83, 4.16) | <0.001 | 3.57(1.25, 10.17) | 0.017* |
| | >10 screened windows | 5 | 0 (0) | 1 | | 1 | |
| Household head heard malaria message 6 months prior to interview | | | | | | | |
| | Heard malaria message | 3468 | 1141 (32.9) | [Reference] | | [Reference] | |
| | Did not hear malaria message | 79 | 36 (45.6) | 1.71 (1.09, 2.67) | 0.019 | 2.01 (0.87, 4.64) | 0.104 |
| Household has ITN access (one ITN for every two household members) | | | | | | | |

*(Continued)*

**Table 11.** (Continued)

| Characteristics | Categories | defacto population | People that did not sleep under ITN, n(%) | Crude OR (95% CI) | P value | *AOR (95% CI) | P value |
|---|---|---|---|---|---|---|---|
| | Has access | 2181 | 600 (27.5) | [Reference] | | [Reference] | |
| | Lack access | 1366 | 577 (42.2) | 1.93 (1.67, 2.22) | <0.001 | 1.80 (1.31, 2.47) | <0.001* |
| Total defacto population with at least one ITN | | 3547 | 1177 (33.2) | | | | |

OR: Odds ratio; AOR: Adjusted odds ratio.

* Statistically associated (AOR) with an increase in not using ITNs the night prior to the study.

that the household has enough ITNs such that every two household members could use one ITN if they decide to do so (access). Given that 86.8% of study population and 65.8% of study households had access to ITN suggests a need to improve the existing strategies of delivering ITNs to the study communities to ensure that all households have at least one ITN for every two household members [8].

**Insecticide treated bed net use.** Across the study communities, an average of 65.1% of the study population used ITN the night before the survey. Among households with at least one ITN for every two household members, 72.5% of their household population used ITN the night prior to the survey.

**Table 12. Summary of sandflies caught in study communities by sex, place of collection, and collection methods.**

| Study Community | Outdoor/indoor collection | Place of collection | Sample collection Method | Sex of sand flies | No. of Flies |
|---|---|---|---|---|---|
| Ashiabre | Outdoor | Household Compound | CDC Light Trap | Female | 77 |
| Ashiabre | Outdoor | Church compound | CDC Light Trap | Female | 2 |
| Ashiabre | Outdoor | School compound | CDC Light Trap | Female | 9 |
| Ashiabre | Outdoor | Household Compound | CDC Light Trap | Male | 14 |
| Ashiabre | Indoor | School Junior High Classroom | Aspiration | Female | 9 |
| Ashiabre | Indoor | School KG Classroom | Aspiration | Female | 2 |
| Ashiabre | Indoor | School Primary Classroom | Aspiration | Female | 3 |
| Ashiabre | Indoor | Inside church | Aspiration | Female | 3 |
| Keri | Outdoor | Household compound | CDC Light Trap | Female | 50 |
| Keri | Outdoor | Church compound | CDC Light Trap | Female | 3 |
| Keri | Outdoor | Mosque compound | CDC Light Trap | Female | 2 |
| Keri | Outdoor | School compound | CDC Light Trap | Female | 8 |
| Keri | Outdoor | Mosque compound | CDC Light Trap | Male | 1 |
| Keri | Outdoor | Household compound | CDC Light Trap | Male | 7 |
| Keri | Indoor | Household sleeping area* | Aspiration | Female | 10 |
| Keri | Indoor | Household sleeping area** | Aspiration | Female | 1 |
| Keri | Indoor | Household sleeping area* | Aspiration | Male | 2 |
| Sibi Hilltop | Outdoor | Household compound | CDC Light Trap | Female | 4 |
| Sibi Hilltop | Outdoor | Church compound | CDC Light Trap | Female | 2 |
| Sibi Hilltop | Outdoor | School compound | CDC Light Trap | Female | 8 |
| Sibi Hilltop | Outdoor | Household compound | CDC Light Trap | Male | 1 |
| **Total** | | | | | 218 |

*Room without bednet.

** Room with old bednet (>6 years).

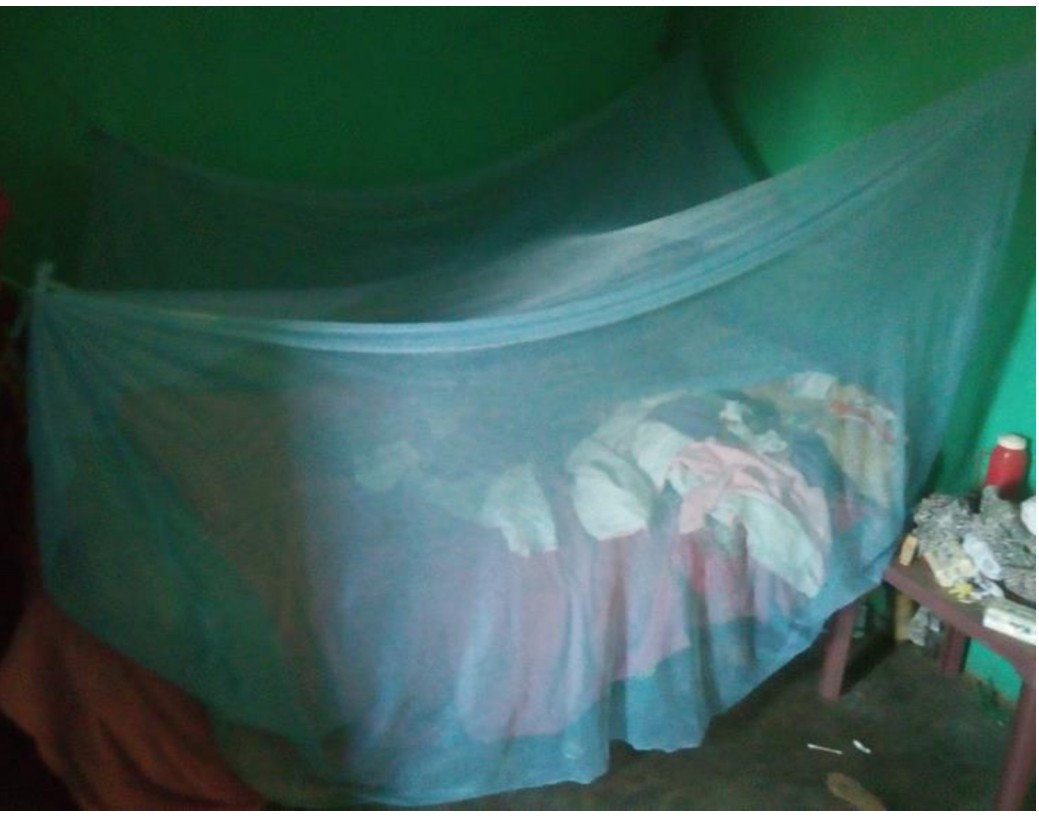

**Fig 1. ITN observed hanging in sleeping area of a household.**

According to the 2014 DHS, only 36% of the household population in Ghana slept under an ITN the night before the survey. In that survey, the then Volta region recorded the highest proportions of household population using the ITN compared to other regions with 53.7% of the household population in the region reported to have slept under an ITN while 64.9% of the household population in households having at least one ITN slept under an ITN the night before the survey [44]. The 2019 GMIS also reported that 54.3% of the household population in the Volta region slept in an ITN while 61.8% of the household population in households having at least one ITN slept under an ITN the night before the survey [43].

As a result, while data obtained from this study indicate a need to improve household access to ITN, it also suggests a need to put in measures to understand why some people in households with household access to ITN fail to use the ITN. This may call for the development of context specific change communications strategy to promote ITN use among the general study population [10, 45].

As a result, universal coverage, was not fully achieved in any of the study communities. However, it is worth mentioning that two indicators, the proportion of households that own at least one ITN, and the proportion of the population with access to an ITN within the household, were above 80%. A recent study has indicated that the attainment of 80% of households owning one ITN per every two household members in a national survey may not be realistic and advocated for the consideration of population access to ITNs as the better indicator of "universal coverage," given that it is based on people as the unit of analysis [8, 46].

The fact that universal coverage of ITN (per the current accepted definition) was not achieved in the study communities suggests a need to review the current ITN delivery

strategies to ensure that all households in these malaria endemic communities of Ghana attain it. This is particularly important because of documented benefits of community wide high coverage of ITN on reduction of malaria morbidity and mortality as well as the anticipated benefits of this intervention against vectors of leishmaniasis in the study communities [46].

Furthermore, there may be a need to investigate what other uses the bed nets may be put to in the study communities which may reduce the number of bed nets available for the household use (Fig 1). This is because 84.5% of household heads indicated that at least one ITN had been disposed of from their households in the 12 months prior to this study. The methods for net disposal indicated include burning, dumping at garbage or refuse dump, or reusing for other purposes.

The mode of disposal of the ITNs and their non-biodegradable packaging materials is of concern due to its potential for environmental and human health harm. Efforts towards recycling these used bed nets should be explored to reduce their potential for human and environmental harm [44].

In addition, majority (81.5%) of those who disposed of at least one bed net indicted that the nets were torn. This observation was similar to what was observed in the 2014 Ghana demographic and health survey in which 82.9% of households surveyed in Ghana indicated that their main reason for treated bed net disposal was because the nets were torn [44].

**Factors associated with failure to use ITN.** In a cross-sectional study conducted in south west Ethiopia, household having decreased access to ITN, and having household size of 4–6 members were significantly associated with failure to use ITN [41]. Another cross-sectional survey conducted in Yemen found having three or more damaged LLINs in the house, individuals aged 16 years and above, and living in huts to be significantly associated with failure to use ITN [40].

Given that in this study, lack of household access to ITN was significantly associated with failure to use it, the national malaria control program should review its delivery mechanisms to ensure that all households own an ITN and at least 80% of all households in the study communities have a minimum of one ITN for every two household members [2, 47].

Having increased family size (>10 members) and more than two screened windows were also observed to be associated with failure to use the ITN in this study. As a result, further studies on sleeping arrangements, housing conditions as well as reasons for non-use of ITN by larger families and households having at least two screened windows may help to develop strategies to improve use of ITNs in the study communities [41].

Factors associated with use of ITNs tend to be context specific and varied. Identifying such factors and the people not using the ITNs presents an opportunity to both explore and understand their reasons for non-use in order to develop and adapt implementation strategies to encourage an increased use of the ITNs among community members [36, 38, 48–50].

Some recent studies in Ghana have concluded that ITN use among persons in households having ITN access is affected by household characteristics and is also spatially dependent. As a result, they advocate for studies that focus on rural settings, urban settings, and wealth status independently to better understand the predictors on ITN use among this group. Additionally, opportunities for improving ITN communication messages has been advocated to improve net use among persons with ITN access in Ghana [51, 52].

**Presence of sand flies within study communities.** Using indoor aspiration and CDC light traps, sand flies were trapped from various locations of the study communities where humans could be found such as homes, churches, mosques, and schools.

Among the vectors of leishmaniasis in the Old World, some phlebotomine sand fly species have been more associated with certain species of the *Leishmania* parasite. The *Leishmania* parasite has more than twenty parasite species known to infect humans. Of about 500 known

phlebotomine sand fly species, only about 30 are known to transmit *Leishmania* parasites [53, 54].

There is therefore a need to confirm the presence of sand flies in an area prior to proceeding with the next steps of investigating the sand fly species as well as determining whether the sand flies observed are infected with *Leishmania* parasites. Identification of sand flies in the communities investigated in this study presents an opportunity for the next steps of sand fly species identification and investigation of *Leishmania* infection to be carried out.

Previous vector studies in Ghana aimed at identifying phlebotomine sand flies have resulted in detecting several species of sand flies mostly belonging to the genus *Sergentymyia* with only two species belonging to the genera *Phlebotomus* (*P. duboscqi and P. rodhaini*) [55].

Some of the previous vector studies in Ghana also confirmed DNA of *Leishmania* parasites such as *Leishmania tropica* and *Leishmania major* in sand flies belonging to the genus *Sergentymyia* [56]. Sand flies belonging to the genus *Sergentomyia* have not been confirmed as vectors of human Leishmaniasis. However, detecting species of *Leishmania* known to cause human leishmaniasis in the Volta Region of Ghana calls for more studies to identify the likely putative vectors of those parasites [24].

Further studies are required to investigate the role of ITN roll out in these study communities on the absence of sand flies observed in majority of the household sleeping rooms having ITN which were selected for indoor aspiration. This is important because the preferred feeding or resting habit of sandflies is also known to influence their usual location. For instance, endophagic sand flies bite indoors whilst exophagic ones bite outdoors. Also, there are sand flies which prefer to rest indoors (endophilic) whilst others prefer to rest outdoors (exophilic) [57, 58].

Detection of sand flies in areas outside the household sleeping areas but with proximity to human activities such as the household compounds, school compounds and classroom calls for a more integrated vector control approach. This will ensure a reduced contact with the sand flies while additional studies are conducted to describe species and vector competence of sand flies in the study communities to transmit leishmaniasis [53, 54, 59].

## Conclusions

Universal coverage for ITN has not been achieved in the study areas. Factors associated with non-use of ITNs such as lack of household access to ITN and having family size of more than 10 members need to be prioritized in future efforts aimed at improving ITN use in the study area. Absence of sand flies in all sleeping areas having a recent ITN and detection of sand flies outside sleeping areas suggest a need for an integrated vector control approach against sand flies in the study area.

## Limitations of the study

Inclusion of a household in the study depended on the consent of the household head. This may have led to the exclusion of a few households, given that 587 households were included out of 600 households invited. Recall bias in terms of response to questionnaire could not be ruled out. Also, investigation of sand fly species and *Leishmania* infection in the sand flies caught would have enriched the data.

## Supporting information

**S1 File. Additional household characteristics.**
(DOCX)

**S2 File. STROBE checklist: Checklist according to the strengthening the reporting of observational studies in epidemiology (STROBE) guidelines.**
(DOCX)

**S3 File. Study questionnaire.**
(PDF)

## Acknowledgments

The assistance of field workers from the Nkwanta South and North District Health directorates is also appreciated. The authors are also grateful for the support from Mr. Emmanuel Agbodogli of the Nkwanta South District hospital and Dr. Laud Boateng, the Nkwanta South District Director during the field component of this study. The logistical and technical support of staff of the U.S Naval medical research unit #3 Ghana detachment during the conduct of this study is also appreciated. We are also grateful to Mr. Sylvester Nyarko and Mr. Mba-Tihssommah Mosore for their support with the sandfly collections.

## Author Contributions

**Conceptualization:** Richard Akuffo, Michael Wilson, Bismark Sarfo, Francis Anto.

**Data curation:** Richard Akuffo.

**Formal analysis:** Richard Akuffo.

**Funding acquisition:** Richard Akuffo, Phyllis Dako-Gyeke, Richard Adanu.

**Investigation:** Richard Akuffo.

**Methodology:** Richard Akuffo, Michael Wilson, Bismark Sarfo, Phyllis Dako-Gyeke, Richard Adanu, Francis Anto.

**Project administration:** Richard Akuffo.

**Resources:** Richard Akuffo.

**Software:** Richard Akuffo.

**Supervision:** Michael Wilson, Bismark Sarfo, Francis Anto.

**Validation:** Richard Akuffo, Michael Wilson, Phyllis Dako-Gyeke, Richard Adanu.

**Writing – original draft:** Richard Akuffo.

**Writing – review & editing:** Richard Akuffo, Michael Wilson, Bismark Sarfo, Phyllis Dako-Gyeke, Richard Adanu, Francis Anto.

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
