## [Decision Letter · Decision Letter 0]

5 Oct 2021

PONE-D-21-25541

Insecticide-treated net (ITN) use, factors associated with non-use of ITNs, and occurrence of sand flies in three communities with reported cases of cutaneous leishmaniasis in Ghana

PLOS ONE

Dear Dr. Richard Akuffo,

Thank you for submitting your manuscript to PLOS ONE. After careful consideration, we feel that it has merit but does not fully meet PLOS ONE’s publication criteria as it currently stands. Therefore, we invite you to submit a revised version of the manuscript that addresses the points raised during the review process.

We look forward to receiving your revised manuscript.

Kind regards,

Alireza Badirzadeh

Academic Editor

PLOS ONE

Journal Requirements:

2. Please include additional information regarding the survey or questionnaire used in the study and ensure that you have provided sufficient details that others could replicate the analyses. For instance, if you developed a questionnaire as part of this study and it is not under a copyright more restrictive than CC-BY, please include a copy, in both the original language and English, as Supporting Information

Reviewers' comments:

Reviewer's Responses to Questions

**Comments to the Author**

1. Is the manuscript technically sound, and do the data support the conclusions?

Reviewer #1: Yes

Reviewer #2: Yes

2. Has the statistical analysis been performed appropriately and rigorously? 

Reviewer #1: Yes

Reviewer #2: Yes

3. Have the authors made all data underlying the findings in their manuscript fully available?

Reviewer #1: Yes

Reviewer #2: Yes

4. Is the manuscript presented in an intelligible fashion and written in standard English?

Reviewer #1: Yes

Reviewer #2: Yes

5. Review Comments to the Author

Reviewer #1: Dear Author

- It is better to add more detail of Leishmania species in the introduction.

-More references on the role of leishmania inside the MQ need to be include in the Introduction section. I suggest the following references:

1- PMID: 28828327

2- PMID: 32595711

-Please add a map of region

- Please interpret the study vividly and strongly.

Reviewer #2: Dear authors;

First of all, I should congratulate you due to carry out this valuable study. But in my opinion, some improvements are necessary to enrich the manuscript. So, I have listed my comments.

1. Add climate status of study area, briefly

2. Explain the importance and status of leishmaniasis in the study area

3. Identify the species or species of collected sandflies (Or explain way it is your limitation!)

4. If possible, determine the association between existence of a CL history in family and the factors that you indicated, such as the use of mosquito nets in the family, number of ITN per person in family, duration of use of ITN… and…

5. After applying comment #4 you can discuss about possible reasons of the results

6. Using the following article, you can add the different aspects of CL in the Middle East, as one of the most important foci of this disease, briefly: Super Infection of Cutaneous Leishmaniasis Caused by Leishmania major and L. tropica to Crithidia fasciculata in Shiraz, Iran.

Best regards

6. PLOS authors have the option to publish the peer review history of their article (what does this mean?). If published, this will include your full peer review and any attached files.

Reviewer #1: No

Reviewer #2: No

---

## [Author Response · Author response to Decision Letter 0]

3 Nov 2021

Journal Requirements:

Response: The manuscript has been revised according to PLOS ONE’s style requirements

2. Please include additional information regarding the survey or questionnaire used in the study and ensure that you have provided sufficient details that others could replicate the analyses. For instance, if you developed a questionnaire as part of this study and it is not under a copyright more restrictive than CC-BY, please include a copy, in both the original language and English, as Supporting Information

Response: Study questionnaire has been included as a supporting information

3. We note that part of your Fig 1 is from "OpenStreetMap and other contributors" and under the CC-BY-SA license (as seen in the bottom right corner). Unfortunately, we cannot publish maps under the CC-BY-SA license since it conflicts with the CC-BY-4.0 license that we use. PLOS ONE publishes all material under the Creative Commons Attribution (CC BY) 4.0 license, which means that they will be freely available online, and any third party is permitted to access, download, copy, distribute, and use these materials in any way, even commercially, with proper attribution.

Response: The previous Fig 1 has been removed. This is to ensure that there is no restriction to making this research output freely available to all who may desire to have access to it. The issues raised regarding copyright have been well noted for subsequent publications. Thank you. 

Reviewer #1: 

Comment: 

Dear Author,- It is better to add more detail of Leishmania species in the introduction.

-More references on the role of leishmania inside the MQ need to be include in the Introduction section. I suggest the following references:

1- PMID: 28828327

2- PMID: 32595711

Response: Reference PMID:28828327 and other references have been incorporated at lines 90-92

Comment: Please add a map of region

Response: A map has been included to illustrate the study area

Comment- Please interpret the study vividly and strongly.

Response: Current interpretation of the study is considered vivid

Reviewer #2: 

Dear authors;

First of all, I should congratulate you due to carry out this valuable study. But in my opinion, some improvements are necessary to enrich the manuscript. So, I have listed my comments.

Comment: Add climate status of study area, briefly

Response: Climate of Ghana and the study area for that matter is provided at line 139 and 140.

Comment: Explain the importance and status of leishmaniasis in the study area

Response: Information provided at line 104-107

Comment: Identify the species or species of collected sandflies (Or explain way it is your limitation!)

Response: The species of sand flies collected was undetermined at the time of writing this publication. The main reason for not detecting the species was due to limited expertise in sandfly species identification among the authors. Also, we had limited funding to hire an expert to conduct the identification. Fortunately, however, we have received assurance of collaboration to investigate the species of the sandflies at the Instituto de salud carlos III in Spain. Once the species of the sand flies is confirmed, the scientific community shall duly be informed.

Comment: If possible, determine the association between existence of a CL history in family and the factors that you indicated, such as the use of mosquito nets in the family, number of ITN per person in family, duration of use of ITN… and…

Response: No association was observed between CL history and the ITN indicators. This may suggest a complex epidemiology of CL in the study area. The results section has accordingly been updated.

Comment: After applying comment #4 you can discuss about possible reasons of the results

Response: This may suggest a complex epidemiology of CL in the study area. The possible reasons seem very varied. I however stated the finding in the results section. I believe additional context specific studies are required to understand the role of the lack of association between family history of CL or exposure to Leishmania infection measured by LST, and increased odds of not sleeping in the ITN. 

Comment: Using the following article, you can add the different aspects of CL in the Middle East, as one of the most important foci of this disease, briefly: Super Infection of Cutaneous Leishmaniasis Caused by Leishmania major and L. tropica to Crithidia fasciculata in Shiraz, Iran.

Response: This is very important. However, I believe we shall find a basis for discussing this peculiarities when the species of Leishmamia in the study area has been detected.

---

## [Decision Letter · Decision Letter 1]

25 Nov 2021

Insecticide-treated net (ITN) use, factors associated with non-use of ITNs, and occurrence of sand flies in three communities with reported cases of cutaneous leishmaniasis in Ghana

PONE-D-21-25541R1

Dear Dr. Akuffo,

We’re pleased to inform you that your manuscript has been judged scientifically suitable for publication and will be formally accepted for publication once it meets all outstanding technical requirements.

Kind regards,

Alireza Badirzadeh

Academic Editor

PLOS ONE

Additional Editor Comments (optional):

Reviewers' comments:

Reviewer's Responses to Questions

**Comments to the Author**

1. If the authors have adequately addressed your comments raised in a previous round of review and you feel that this manuscript is now acceptable for publication, you may indicate that here to bypass the “Comments to the Author” section, enter your conflict of interest statement in the “Confidential to Editor” section, and submit your "Accept" recommendation.

Reviewer #1: All comments have been addressed

Reviewer #2: All comments have been addressed

2. Is the manuscript technically sound, and do the data support the conclusions?

Reviewer #1: Yes

Reviewer #2: Yes

3. Has the statistical analysis been performed appropriately and rigorously? 

Reviewer #1: Yes

Reviewer #2: Yes

4. Have the authors made all data underlying the findings in their manuscript fully available?

Reviewer #1: Yes

Reviewer #2: Yes

5. Is the manuscript presented in an intelligible fashion and written in standard English?

Reviewer #1: Yes

Reviewer #2: Yes

6. Review Comments to the Author

Reviewer #1: All changes I requested from the authors in the first draft were done completely. Therefore, I accept the MS and the paper is acceptable for publication in its present form in the Journal

Reviewer #2: Dear authors;

Thanks for your point-to-point reply. Previous correction suggestions are accepted.

Best regards

7. PLOS authors have the option to publish the peer review history of their article (what does this mean?). If published, this will include your full peer review and any attached files.

Reviewer #1: No

Reviewer #2: No

---

## [Editor Report · Acceptance letter]

3 Dec 2021

PONE-D-21-25541R1 

Insecticide-treated net (ITN) use, factors associated with non-use of ITNs, and occurrence of sand flies in three communities with reported cases of cutaneous leishmaniasis in Ghana

Dear Dr. Akuffo:

I'm pleased to inform you that your manuscript has been deemed suitable for publication in PLOS ONE. Congratulations! Your manuscript is now with our production department. 

Kind regards, 

on behalf of

Dr. Alireza Badirzadeh 

Academic Editor

PLOS ONE